# RORγt inverse agonists demonstrating a margin between inhibition of IL-17A and thymocyte apoptosis

Mia Collins[1]*, Rikard Pehrson[2], Hanna Grindebacke[1], Agnes Leffler[1], Marie Ramnegård[1], Helena Rannikmäe[1], Nina Krutrök[1], Linda Yrlid[1], Charlotte Pollard[1], Ian Dainty[1], Frank Narjes[3], Stefan von Berg[3], Antonio Llinas[2], Anna Malmberg[2], Jane McPheat[4], Eva Hansson[4], Elisabeth Bäck[1], Jenny Bernström[4], Thomas G. Hansson[5], David Keeling[5], Johan Jirholt[1]

1 Bioscience COPD/IPF, Research and Early Development, Respiratory & Immunology, BioPharmaceuticals R&D, AstraZeneca, Gothenburg, Sweden, 2 Drug Metabolism and Pharmacokinetics, Research and Early Development, Respiratory & Immunology, BioPharmaceuticals R&D, AstraZeneca, Gothenburg, Sweden, 3 Medicinal Chemistry, Research and Early Development, Respiratory & Immunology, BioPharmaceuticals R&D, AstraZeneca, Gothenburg, Sweden, 4 Discovery Sciences, BioPharmaceuticals R&D, AstraZeneca, Gothenburg, Sweden, 5 Projects, Research and Early Development, Respiratory & Immunology, BioPharmaceuticals R&D, AstraZeneca, Gothenburg, Sweden

* Mia.collins@astrazeneca.com

**Data Availability Statement:** All relevant data are within the manuscript and its Supporting Information files.

## Abstract

Multiple genetic associations suggest a causative relationship between Th17-related genes coding for proteins, such as IL-17A, IL-23 and STAT3, and psoriasis. Further support for this link comes from the findings that neutralizing antibodies directed against IL-17A, IL-17RA and IL-23 are efficacious in diseases like psoriasis, psoriatic arthritis and ankylosing spondylitis. RORγt is a centrally positioned transcription factor driving Th17 polarization and cytokine secretion and modulation of RORγt may thus provide additional benefit to patients. However, RORγt also plays a role in the normal development of T cells in the thymus and genetic disruption of RORγt in the mouse leads to the development of lymphoma originating in the thymus. Whilst it is not established that down-regulation of RORγt activity would lead to the same consequence in humans, further understanding of the thymus effects is desirable to support progress of this target as a potential treatment of Th17-driven disease. Herein we present the characterisation of recently disclosed RORγt inverse agonists demonstrating target engagement and efficacy *in vitro* and *in vivo* against Th17 endpoints but requiring higher concentrations *in vitro* to affect thymocyte apoptosis.

## Introduction

Th17 cells were discovered some 20 years ago [1,2] and rapidly became the focus of intense study. These cells play an important role in host defense against extracellular bacteria and fungal infections caused by pathogens such as mycobacteria and candida [3]. Such actions are mediated, in part, through the production of pro-inflammatory cytokines such as IL-17. The

**Funding:** The author(s) received no specific funding for this work.

**Competing interests:** All authors are or have been employees of AstraZeneca. Several of the authors are or have been shareholders of AstraZeneca. This does not alter our adherence to PLOS ONE policies on sharing data and materials.

differentiation and maturation of Th17 cells is also aided by the cytokine IL-23 which also helps to maintain their pro-inflammatory phenotype [4]. However, when dysregulated, Th17 cells can be important drivers of pathological processes leading to autoimmune diseases [5]. In psoriasis, clinical studies with neutralizing antibodies towards IL-17A and IL-23 have demonstrated efficacy [6–8]. These therapies have also been reported to be efficacious in the treatment of other diseases such as ankylosing spondylitis, psoriatic arthritis [9] and Crohn's disease [10].

Th17 cells are dependent on the transcription factor retinoic acid related-orphan nuclear receptor γt (RORγt) for their polarization and the production of IL-17A [11]. RORγt has in addition been shown to down regulate the expression of FoxP3 and enhance transcription of genes coding for IL-17F, IL-22, IL-21, and IL-23R [12,13].

However, in the mouse thymus, αβ T cell precursors require the expression of RORγt during positive selection and maturation. Upon TCRβ selection, the thymic precursor cell enters a rapid proliferative burst phase. This is followed by the upregulation of RORγt leading to a quiescent state where recombination of the TCRα locus ensues [14,15]. After successful TCRα recombination a fully functional T cell receptor is upregulated on the cell surface, together with both CD4 and CD8, constituting double positive thymocytes. About 85% of thymocytes are double positive and remain dependent on RORγt for their survival at this developmental stage [16–18]. We use this dependency later in this study to evaluate *in vivo* effects of our compounds on thymocytes. In the mouse, thymocytes deficient in RORγt remain hyperproliferative but they also rapidly apoptose through the lack of RORγt-dependent Bcl-xL induced quiescence [17,19]. RORγt is encoded by the *RORC* gene and disruption of the *Rorc* gene in mice has been shown to lead to development of thymic lymphoblastic lymphomas. It is not yet known if this observation is relevant for other species such as humans, as some data suggest species differences in thymus biology [20,21]. However, when considering the development of drugs inhibiting RORγt, it would be important to establish a margin between any thymocyte effects and the desired inhibition of peripheral cytokine production in Th17 cells [22].

The RORC gene produces two isoforms of RORγ, γ and γt, which differ slightly in the N-terminus but share the exons encoding the ligand binding domain. Thus, pharmaceutical intervention directed towards this domain is likely to inhibit both isoforms similarly. The isoforms also differ in their tissue distribution where RORγt is restricted to immune cells while RORγ is expressed in many tissues such as liver and muscle [23]. Interestingly, both RORγ and RORγt are present in Th17 cells whereas only RORγt is present in thymocytes [24]. For simplicity we will refer to both isoforms as RORγt since we do not expect the compounds described here to differentiate between them.

RORγt is also important in other cell types where it drives cytokine expression and polarization [25]. Amongst these are γδ T cells, which are rapid innate-like producers of IL-17A and have been shown to be of importance in human psoriatic skin [26]. Preclinical *in vivo* models of psoriasis and skin inflammation in rodents often use imiquimod (IMQ, a TLR7/8 agonist) which, after application on the skin, gives rise to local inflammation characterised by erythema, swelling, scaling and thickening of the skin similar to psoriasis in the human [27]. IMQ drives a mixed inflammation but it is described to be mediated by the IL-23/IL-17 axis [27] and γδ T cells have been suggested to be the main producers of IL-17A in this model [28].

Whilst antibodies against Th17-related cytokines have proved efficacious against psoriasis and a number of related diseases, small molecule inhibition of this nuclear hormone receptor may be an attractive route for intervention and provide additional benefit to the patient compared with single cytokine neutralization through antibodies and several papers describe efforts towards this end [29,30].

In this paper we describe further pharmacological characterisation of six compounds from our chemical campaign, previously disclosed by us, that are inverse agonists of RORγt [31–33]. The compounds exhibited a robust inhibition of IL-17A release from human primary Th17 cells and show high affinity to the RORγt receptor. Compounds **1**–**4** belong to the acetamide series, from which the clinical candidate AZD0284 (**5**) was eventually derived [33]. Compound **6** is a more potent analogue of **5**. Compound **3** was obtained by separation of the racemic mixture, as has been described for compound **2**. In this study we characterised additional RORγt inverse agonists and investigated their effect on thymocyte apoptosis and inhibition of cytokine release.

## Material and methods

### Screening assays

**Human RORγt radioligand competition binding assay.**   This assay was used to test whether compounds could inhibit the binding of tritiated 2-(4-(ethylsulfonyl)phenyl)-*N*-(4-(2-(methoxymethyl)phenyl)thiophen-2-yl)acetamide to recombinant human RORγt LBD ($pIC_{50}$ = 7.32, n = 6).

The assay was run in white polystyrene flat-bottom 384-well plates (Greiner, cat. No. 781075). Assays were carried out in 40 μl reaction volumes. A concentration range of test ligands in 0.4 μl of DMSO were added to assay plates using an acoustic liquid dispenser. 4 nM purified N-(HN)6-GST-TCS-hRORγ (258–518) was mixed with 1 mgml$^{-1}$ yttrium oxide (YOx) glutathione SPA imaging beads in assay buffer (20 mM Tris, 150 mM NaCl,10% glycerol, 0.25% CHAPS, 1 mM TCEP) prior to adding to 30 μl to test ligands. Assay plates were incubated for 1 hour at room temperature before adding 10 μl tritiated 2-(4-(ethylsulfonyl)phenyl)-*N*-(4-(2-(methoxymethyl)phenyl)thiophen-2-yl)acetamide to test plates in assay buffer (final concentration, 25 nM). Test plates were incubated for 16 hours and read using a LEADseeker (GE Healthcare, USA) Multimodality imaging instrument. $K_i$ was calculated from the $IC_{50}$ value using the equation $K_i = IC_{50}/(1+[L]/Kd)$ where [L] = 25 nmol/l and $K_d$ = 17 nmol/l.

**Murine RORγt radioligand competition binding assay.**   This assay was used to test whether compounds could inhibit the binding of the same tritiated ligand used in the human binding assay described above to recombinant murine RORγt LBD ($pIC_{50}$ = 6.91, n = 3). The assay was run in white/clear polystyrene flat-bottom 384-well plates (Greiner, cat. No. 781095). Assays were carried out in 40 μl reaction volumes. 0.125 μl of a fixed concentration range of test ligands dissolved in DMSO were added to assay plates using an acoustic liquid dispenser. 10nM purified HN-AVI-GST-TCS-mRORγ(aa256-516) was mixed with 0.75 mgml$^{-}$ yttrium silicate (Ysi) glutathione coated beads in assay buffer (20 mM Tris pH7.5, 150 mM NaCl,10% glycerol, 0.25% CHAPS, 1mM DTT) prior to adding 30 μl to test ligands. Assay plates were incubated for 30 minutes at room temperature before adding 10 μl tritiated 2-(4-(ethylsulfonyl)phenyl)-*N*-(4-(2-(methoxymethyl)phenyl)thiophen-2-yl)acetamide to test plates in assay buffer (final concentration, 15 nM). Test plates were incubated for 8–20 hours and read using a 1450 MicroBeta TriLux Liquid Scintillation Counter (Perkin Elmer, MA, USA). $K_i$ was calculated from the $IC_{50}$ value using the equation $Ki = IC_{50}/1+[L]/Kd)$ where [L] = 15 nmol/l and $K_d$ = 45 nmol/l.

**Human RORβ radioligand competition binding assay.**   This assay was used to estimate compound potency for inhibition of radioligand binding ($^3$H- all-trans retinoic acid) to the human retinoic acid receptor-related orphan receptor beta (RORB).

The assay (SPA) was run in white/clear polystyrene flat-bottom 384-well plates (Greiner, cat. No. 781095). Assays were carried out in 50 μl reaction volumes. A 0.5 μl volume of a fixed

concentration range of test ligands dissolved in DMSO were added to assay plates using an acoustic liquid dispenser. Assay buffer (50 mM Tris pH 7.5, 150 mM NaCl, 10% Glycerol, 0.1% Triton X-100, 1 mM TCEP) containing 150 nM 6H-AVI-GST-TEV-hRORβT221-K470) and 0.8 mgml⁻¹ Yttrium silicate (YSi) copper SPA beads in was prepared and 40 μl was added to test ligands. Assay plates were incubated for 30 minutes at room temperature before adding 10 μl tritiated all-trans retinoic acid to test plates in assay buffer (final concentration, 30 nM). Test plates were incubated for 18–23 hours and read using a 1450 MicroBeta TriLux Liquid Scintillation Counter (Perkin Elmer, MA, USA).

**RORγt co-factor recruitment assay.** A co-activator recruitment assay was used to test whether compounds could inhibit the recruitment of steroid receptor coactivator peptide 1 (NCOA1-677-700) to the human retinoic acid receptor-related orphan receptor gamma (RORC) LBD.

The assay was run in black 384 well plates (Greiner cat no: 784900). Various concentrations of test ligands in 0.1 μl DMSO were dispensed to assay plates using an Echo acoustic dispenser (Beckman Coulter, UK). Two pre-mixes were prepared and incubated for 1 hour at room temp in the dark. Pre-mix 1 comprised 100 nM protein (biotinylated HN-Avi-MBP-TCS-hRORγ 258–518)) and 60 nM streptavidin APC in assay buffer, 50 mM MOPS pH7.4, 50 mM KF, 0.003% (w/v) CHAPS, 10 mM DTT and 0.01% (w/v) BSA and pre-mix 2 comprised 160 nM biotinylated SRC-1 peptide (NCOA1-677-700) and 20 nM europium-W8044 labelled Streptavidin in assay buffer. Five μl of pre-mix 2 was dispensed to assay plates containing test compound and incubated for 15 minutes prior to adding 5 μl of pre-mix 1. Plates were incubated at room temperature for 1 hour in the dark, prior to reading in a Pherastar multi-mode plate reader (BMG LABTECH, Ortenberg, Germany) using HTRF filter set (ex 320 nm, em 612 and 665 nm). The FRET signal at 665 nm was divided by the signal at 612 nm and multiplied by 10,000 to generate a signal ratio value for each well.

**RORα co-factor recruitment assay.** The aim of this assay was to show whether compounds could inhibit or stimulate the recruitment of peroxisome proliferator-activated receptor γ coactivator 1-α (PGC1alpha 130–154) peptide to the human RORα LBD in a co-activator recruitment assay. The hypothesis was that if compounds could bind to the receptor they would either inhibit or stimulate recruitment of the PGC1alpha 130–154 coactivator peptide.

The assay was run in black 384 well plates (Greiner: 784900). A 0.2 μl volume of a fixed concentration range of test ligands dissolved in DMSO were added to assay plates using an acoustic liquid dispenser. Two pre-mixes were prepared in falcon tubes. Pre-mix 1 comprised 180 nM Protein (His6-tcs-hRORαLBD) and 8 nM Lance Eu-W1024-anti 6xHis in assay buffer, 50 mM Hepes pH7.4, 100 mM NaCl, 0.1% (w/v) BSA, 1 mM TCEP which was incubated for 30 minutes at room temperature before dispensing 10 μl to assay plates. Then, pre-mix 2 comprising 200 nM biotinylated PGC1alpha peptide and 50 nM streptavidin APC in assay buffer, was prepared and incubated for 30 minutes at room temperature before adding 10 μl to assay plates containing test compound and premix 1. Plates were incubated at room temperature for 1 hour in the dark, prior to reading in a Pherastar multi-mode plate reader (BMG LABTECH) using HTRF filter set (ex 320 nm, em 612 and 665 nm). The FRET signal at 665 nm was divided by the signal at 612 nm and multiplied by 10,000 to generate a signal ratio value for each well.

## Biological assays

**Primary human Th17 cell assay.** Peripheral blood mononuclear cells were isolated from heparin treated human whole blood from healthy donors by density gradient centrifugation. Th17 cells (CD4⁺CXCR3⁻CCR6⁺) were enriched using a human Th17 Cell Enrichment Kit

(cat #18162, STEMCELL Technologies, Cambridge, UK) according to the manufacturer's protocol. The isolated Th17 cells were activated with anti-CD3, anti-CD28, and anti-CD2 beads (Miltenyi Biotech, Bergisch Gladbach, Germany) and cultured in X-Vivo15 medium (Lonza, Basel, Switzerland) supplemented with L-glutamine, β-mercaptoethanol (Gibco/Invitrogen, Thermo Fisher Scientific, MA, USA) and a cytokine cocktail consisting of; 10 ng/ml IL-2 (Gibco/Invitrogen), 20 ng/ml IL-6 (PeproTech Nordic, Stockholm, Sweden), 100 ng/ml IL-23, 20 ng/ml IL-1β, and 2 ng/ml TGF-β (R&D Systems Inc, MN, USA). Cells were seeded at 8000 cells/well in a 384-plate (cat #3707, Corning, MA, USA) in the presence of compounds or DMSO and cultured for 4 days (37˚C, 5% $CO_2$). On day 4, supernatants were collected and IL-17A was measured using a Human IL-17A HTRF Assay kit (Cisbio/PerkinElmer, MA, USA) according to the manufacturer's protocol.

In addition, the cytotoxic effect of compounds was evaluated on the cell plates (after removal of cell media supernatants for IL-17A analysis) using CellTiter-Glo® Luminescent Cell Viability assay (Promega Biotech AB, Nacka, Sweden) that measures adenosine triphosphate (ATP) levels which reflects the number of metabolically active cells.

**Mouse thymocyte apoptosis assay *ex vivo*.**   The thymus from a female C57BL/6NCrl (Charles River) mouse aged around 7 weeks was dissected out after giving the mouse a lethal *i. p*. injection of penthobarbital. Single cell suspension of thymocytes was prepared by disrupting the thymus through a 70 μm cell strainer with the top of a sterile 5 ml syringe plunger in DPBS (without Ca/Mg). Cells were filtered through a 40 μm cell strainer and washed in serum-free media before being resuspended in complete media (RPMI1640 with L-glutamine supplemented with 10% heat inactivated foetal bovine serum (FBS) (Gibco/Invitrogen) and 50 μM 2-mercaptoethanol (Gibco/Invitrogen).

Cells were seeded at 500 000 cells/well in 96-well plates (U-bottom, cat #3799, Corning, MA, USA) in the presence of compounds or DMSO and cultured for 26 hours (37˚C, 5% $CO_2$). The thymocytes were then washed twice in cold DPBS (without Ca/Mg) and stained with APC AnnexinV (#550475, BD Pharmingen, BD Biosciences, CA, USA) and 7AAD (#559925, BD Pharmingen) in AnnexinV Binding buffer (#556454, BD Pharmingen) for 15 minutes at RT (dark) in order to separate live cells from dead, early apoptotic and late apoptotic cells when analysed on the BD Accuri C6 (BD Biosciences, CA, USA).

The compound effect on the percentage of surviving cells relative to the DMSO control were calculated using the following formula;

% Relative survival = (% viable cells / mean % viable cells of DMSO controls) x 100 and plotted as dose-response curves.

**Thymus involution *in vivo*.**   Female C57BL/6NCrl mice (Charles River, MA, USA) arrived at AstraZeneca at an age of 5 weeks +/- 2 days and were organized into groups and acclimatized for one week in order to reduce stress and variability of thymus weight. The mice in the experiment were intentionally young to reduce the impact of the normal process of thymic involution that starts at 4 weeks as this introduces further variability with regards to thymus size and cellularity. Additionally, age-related thymic involution is at its largest post puberty and less pronounced in females [34]. Prior to start of the experiment, the mice were weighed, and tail marked for identification. Two separate studies were conducted. For compound **4**, three doses were selected (15, 50 and 150mg/kg) with a control group dosed with the vehicle (0.5% w/w HPMC and 0.5% w/w F127) (n = 10 in each group). For compound **3**, 50mg/kg (n = 8) and 150mg/kg (n = 4) were selected, with a control group dosed with an inactive compound from the same series (5 mg/ml HPMC, 1 mg/ml Tween). Animals were dosed twice daily perorally at 7.30 and 15.30 except for the 150mg/kg group for compound **3** which got only one dose at 7.30 the final day (in total 3 doses during the treatment period).

Mice were terminated 48 hours post first dose via anesthetization with isoflourane followed by carbon dioxide asphyxiation. Thymi were dissected, weighed and stored on ice in PBS until all tissues were harvested and could proceed immediately to further analysis.

**IMQ induced skin inflammation *in vivo*.** Male C57BL/6NCrl mice (Charles River, MA, USA) arrived at an age of 7 weeks +/- 2 days and were randomly organized into cages marked with an identifier. Prior to start of the experiment the mice were weighed to facilitate assessment of general health post disease induction and during treatment. After disease induction mice were weighed at day 1, 5 and 8. The study included 2 groups: Control group (n = 12), dosed with a non-active compound from the same series and a group dosed with compound **3** at 50 mg/kg (n = 8). Animals were dosed perorally at 07:30 and 15:30 for 7 days. Both experimental groups were treated on the inner and outer side of both ear pinnas with Aldara (IMQ) (Meda, Solna, Sweden) cream daily for 7 days and terminated 24 hours later. The application of Aldara cream took place in the morning after oral compound dosing. The level of ear inflammation (redness, swelling and thickness of the ear) was assessed at day 0 and day 4–7. A micrometer screw with a 4 mm diameter piston tensioned at 2 N was used to measure the ear thickness and a total of 2 measurements per ear were taken and the average was logged. At the day of termination, animals were terminated by cervical dislocation under isoflourane anesthesia, death was confirmed through cardiac laceration. The ear thickness was measured where after ears were cut off and the inner and outer skin were separated and put in PBS (without Ca/Mg) until the preparation for flow cytometric analysis.

**Cell preparations from tissues.** For the thymus involution model, thymi were disrupted and run through a cell strainer. Red blood cells were lysed using Pharmlyse (555899) (BD) and cells were filtered and washed after which they were resuspended in 1 ml of PBS with 2% FBS (Gibco/Invitrogen, Thermo Fisher Scientific, MA, USA). A small aliquot was taken from each sample for cell counting (Sysmex, Kobe Japan).

For the IMQ model, ears were digested for 1.5 hours using Liberase TL research grade (5401020001) (Roche, Basel, Switzerland) and processed using the OctaMACS (Miltenyi) program C. Ear cells were filtered and washed, after which they were resuspended in 1 mL of PBS containing 5% FBS (Gibco/Invitrogen). An aliquot (100 µl) was taken off from the ear cell suspension for CD45 cell counting using the count bright beads (C36950) (Life Technologies, CA, USA). 450 µl were used for phenotypic analysis of cell populations in the ear and the rest was stimulated for 3.5 hours with Leukocyte activation cocktail (550583) (BD) and then stained for flow cytometric analysis.

**Preparation for flow cytometric analysis.** For the thymus involution model, approximately 2 million cells were used for flow cytometric staining. Cells were washed and stained with a viability dye (L34975) (Live/Dead aqua, Life Technologies) and non-specific binding was blocked by using FC-block (553141) (BD) after which extracellular staining for flow cytometric analysis was performed using antibodies directed against surface antigens CD4 V450 (RM4-5) (560468) (BD) and CD8 PE-Cy7 (53–6.7) (561097) (BD Pharmingen) incubating for 30 minutes at 4˚C. Cells were washed and analysed on the BD FACS Fortessa (BD).

For the IMQ model, cells were washed and stained with a viability dye (Live/Dead aqua, Life Technologies) and non-specific binding was blocked by using FC-block (BD) after which extracellular staining for flow cytometric analysis was performed using antibodies directed against surface antigens CD45 FITC (30-F11) (553080) (BD), CD4 APC (RM4-5) (553051) (BD), TCRγδ PE-Cy7 (GL3) (25-5711-80) (eBioscience, CA, USA), CD3 Alexa Fluor 700 (500A2) (557984) (BD) and TCRβ APC-Cy7 (H57-597) (560656) (BD Pharmingen). Cells were incubated for 30 minutes at 4˚C and then washed and fixed for 1 hour in room temperature using the FoxP3 transcription factor staining buffer set (cat# 00-5523-00) (eBioscience). Cells were washed with permeabilization buffer and FC blocked in room temperature for 15

minutes where after intracellular staining was performed using antibodies directed against IL-17A BV421 (TC11-18H10) (563354) (BD), INFg PE (XMG1.2) (554412) (BD). Cells were incubated at 4˚C for 45 minutes and then washed with permeabilization buffer and analysed on the BD FACS Fortessa (BD).

**Analysis parameters.** Thymocytes were gated on their forward and side scatter (FSC/SSC) properties and single cells were selected. The parameter "Time" was used to detect any variances in the flow. Live cells were selected after which CD4 and CD8 were plotted against each other. Double positive thymocytes (expressing both CD4 and CD8 on their surface) were gated. The absolute number of double positive thymocytes were calculated from the flow cytometry data for live double positive thymocytes and the white blood cell count from the Sysmex and presented as absolute number of double positive thymocytes.

Ear cells were gated on their FCS/SSC properties and single cells were selected. The parameter "Time" was used to detect any variances in the flow. Live cells were selected after which the leukocyte marker CD45 was gated. TCRγδ intermediate expressing cells that were negative for the TCRβ marker were gated and IL-17A was selected (flow cytometric gating presented in S1 Fig). IL-17A producing γδ T cells in the ear at day 8 was the primary readout and was evaluated using GraphPad Prism, US v5.0 (GraphPad, CA, USA). All flow cytometric analysis was performed using FlowJo software (FlowJo, OR, USA).

## Statistical evaluation

The raw data from the radio-ligand and co-factor recruitment assays were transformed to % effect using the equation:

Compound % effect = 100*[(X-min)/(max-min)]

where X represents the effect in the presence of test compound and max and min are the effects in presence of the maximum and minimum controls respectively.

The data was plotted to generate concentration response profiles and the concentration-response curves were fitted to the data using the non-linear regression analysis; 4 parameter logistic smart fit method in the Analyser application of the Genedata$^®$ Screener software (Genedata, Inc., Basel, Switzerland). The $pIC_{50}$ and $pEC_{50}$ values were calculated as the negative logarithm of the molar concentration of compound required for 50% inhibition or stimulation in measured effect. $K_i$ was calculated from the $IC_{50}$ value using the equation $Ki = IC_{50}/1 +[L]/Kd)$

Genedata Screener®️ software (Genedata, Inc., Basel, Switzerland) was used for data normalization, curve fitting and calculation of $IC_{50}$ for IL-17A inhibition as well as for cell viability. Raw data from the Human IL-17A HTRF Assay and CellTiter-Glo®️ Luminescent Cell Viability assay was transformed to % effect according to the formula:

Compound % effect = 100*[(X-min)/(max-min)],

where X represents the effect in the presence of test compound. For IL-17A calculations, DMSO was used as minimum (min) inhibition control and RORγt inverse agonist 3-(1,3-benzodioxol-5-yl)-1-(3,5-dimethylpiperidin-1-yl)-3-(2-hydroxy-4,6-dimethoxyphenyl)propan-1-one) at 10 μM was used as the maximum (max) inhibition control [35,36]. For the cell viability calculations, 2-(2,2-dicyclohexylethyl)piperidine at 10 μM was used as maximal toxicity control while DMSO was used as minimal toxicity control. A four-parameter logistic smart fit method in the Analyser application of the Genedata$^®$ Screener software was used for curve fitting of the % effect data and $IC_{50}$ calculation. If a compound reduced the cell viability more than 30% in the CellTiter-Glo 2.0 viability assay at a certain concentration, the corresponding data point for IL-17A production was excluded from the curve fitting.

The IC$_{50}$ values for the tested compounds in the thymocyte apoptosis assay *ex vivo* were determined in GraphPad Prism by fitting the % relative survival data in a four-parameter logistic curve model.

Flow cytometric data for double positive thymocytes was analysed using one-way ANOVA followed by Sidak's post-test for multiple comparisons. Cell populations in the ear of IMQ treated mice were analysed using unpaired t-test. Ear thickness data was analysed using two-way ANOVA with Bonferroni post-test, error bars indicate SEM. Box and whiskers graphs are shown with error bars indicating min-max. Significance levels are used as follows: * ($p \leq 0.05$), ** ($p \leq 0.01$), *** ($p \leq 0.001$).

Dose-response curves for the different comparisons were generated with non-linear regression analysis with log agonist vs response, four parameters, variable slope model.

## Pharmacokinetics and pharmacodynamics

*In vitro.* For the comparisons of effect size across assays, the normalized effect size was plotted for the corresponding concentrations. Where the exact concentration did not correspond between the assays an effect size was extrapolated for that concentration for one of the assays based on the IC$_{50}$ curve fit.

**In vivo.** Test compound concentrations were analysed in blood obtained from the vena saphena by means of capillary tubes and analysed by LC-MS/MS. The terminal concentrations obtained were merged with pharmacokinetic data from satellite animals (animals dosed with compound but not used for other readouts) using the same formulation and dose route. By means of a one compartment model using a population approach (Phoenix®WinNonLin® Certara L.P. Build 6.4.0.768, Princeton, NJ, USA), the individual exposure (AUC) up to termination was estimated. The average concentration was calculated by dividing with the treatment length (48 hours).

## Ethics statement

Whole blood was supplied from healthy AstraZeneca blood donors under written consent approved by the Ethics committee in Gothenburg, Sweden, approval Dnr T705-14 Ad 033–10 during the period 2014-08-25 until 2017-01-16.

All animal experiments were approved by the Gothenburg Ethics Committee for Experimental Animals in Sweden and conform to ethical license No. 47–2013 and No. 142–2015. The approved site number is 31-5373/11 for the animal facilities and all efforts were made to minimize suffering.

## Results

During the development of RORγt inhibitors, we set out to perform a more extensive evaluation of the pharmacological properties of selected compounds. The chemical evolution and characterisation as RORγt inhibitors, has been published recently [31–33]. Compound structures can be found in S2 Fig.

### Binding assays

Firstly, the propensity of the compounds to compete with a known RORγt ligand for binding to the RORγt LBD was evaluated. All compounds were able to inhibit the binding of the tritiated RORγt ligand to the human RORγt LBD in competition binding studies with sub micromolar potency (Table 1). The compounds were then tested for their ability to displace a RORγt

**Table 1. Overview of compound potencies in multiple assays.**

|  | 1 | 2 | 3 | 4 | 5 | 6 |
|---|---|---|---|---|---|---|
| Binding Human RORγt | 8.1±0.1 (5) | 6.6 ± 0.1 (4) | 6.7 ± 0.1 (3) | 6.5 ± 0.0 (3) | 7.1 ± 0.1 (11) | 8.1 (1) |
| Binding Murine RORγt | 7.5 ±0.1 (2) | 6.5 (1) | 6.4 ± 0.1 (2) | 6.3 ± 0.1 (3) | 7.1 ± 0.1 (3) | 7.7 (1) |
| Binding RORβ | n.d. | <5.0 (3) | <5.0 (3) | <5.0 (4) | <5.0 (6) | <5 (2) |
| Inverse agonism RORα | 4.7 (1) | <4.5 (1) | <4.5 (2) | <4.5 (2) | <4.5 (1) | <4.5 (1) |
| Agonism RORα | <4.5 (1) | 5.6 (1) | 5.3 ± 0.1 (2) | <4.5 (2) | 6.4 (1) | 6.5 (1) |
| Inverse agonism RORγt | 7.2 (1) | 7.4 ± 0.1 (2) | 7.3 ± 0.2 (2) | 7.3 ± 0.1 (2) | 7.4 ± 0.1 (11) | 7.5 (1) |
| Thymocyte apoptosis | 6.2 ±0.1 (3) | 5.8 ± 0.1 (3) | 5.6 ± 0.1 (3) | 5.6 ± 0.1 (2) | 6.6 ± 0.1 (4) | 7.5 (2) |
| Human IL-17A in Th17 cells | 7.2 ± 0.2 (5) | 7.3 ± 0.1 (9) | 7.3 ± 0.3 (6) | 7.5 ± 0.3 (6) | 7.8 ± 0.2 (45) | 8.2 ±0.2 (6) |
| Mouse PPB (% unbound) | 0.5 (1) | 10.9 (1) | 6.2 (1) | 17.9 (1) | 18.6 (1) | 5.1 (1) |

[a]The arithmetic mean of $pIC_{50}$ / $pEC_{50}$ values is shown ± standard deviation, where appropriate; in brackets: The number of independent test occasions. N.d.: Not determined. PPB: Plasma protein binding.

ligand from the mouse RORγt LBD. The potencies were similar between the human and mouse binding assays with the largest difference being fourfold (Table 1).

## Co-factor modulation assay

Having established that the compounds bind to RORγt we set out to assess the mode of action through a co-activator recruitment assay. This assay can discriminate agonist, partial agonist and inverse agonist activities of the tested compounds. A clear concentration dependent displacement of the SRC-1 co-activator peptide was achieved for all compounds (Table 1), demonstrating robust inverse agonism with a maximal inhibition of 100%.

## Compound selectivity

RORγt is related to two other proteins, RORα and RORβ, that together constitute the RAR-related orphan receptor family. To evaluate selectivity towards the different ROR isoforms, the compounds were tested in a RORα cofactor modulation assay assessing both inhibitory and agonistic compound effects. If the compounds could bind to the receptor, they would either inhibit or stimulate recruitment of the PGC1alpha 130–154 coactivator peptide.

No inhibitory effect could be detected for any of the compounds up to the highest tested concentration (≤33μM). However, agonism with an EC50 activity <10μM was observed for compounds **2–3** and **5–6** at various maximal PGC1alpha recruitment levels. This agonist activity did not correlate with either potencies or effects in RORγt dependent assays (Table 1). The compounds were further analysed using a radio-ligand binding assay assessing the ability of the compounds to compete for binding with 3H-all-trans retinoic acid to the LBD of the RORβ protein. In this assay no binding could be detected at any concentration tested (≤10μM). These data are summarized in Table 1.

## Biological effects

**Inhibition of IL-17A release from human Th17 cells.** To evaluate if these compounds were efficacious in a human system, we set out to determine the inhibitory effect on IL-17A production in primary human T cells. Briefly, Th17 cells were enriched from blood of healthy donors, activated, and stimulated in a Th17 cytokine milieu with or without test compounds. The cumulative production of IL-17A was measured in the supernatants after 4 days and dose response curves were generated (all dose curves are presented in Fig 1). The compounds

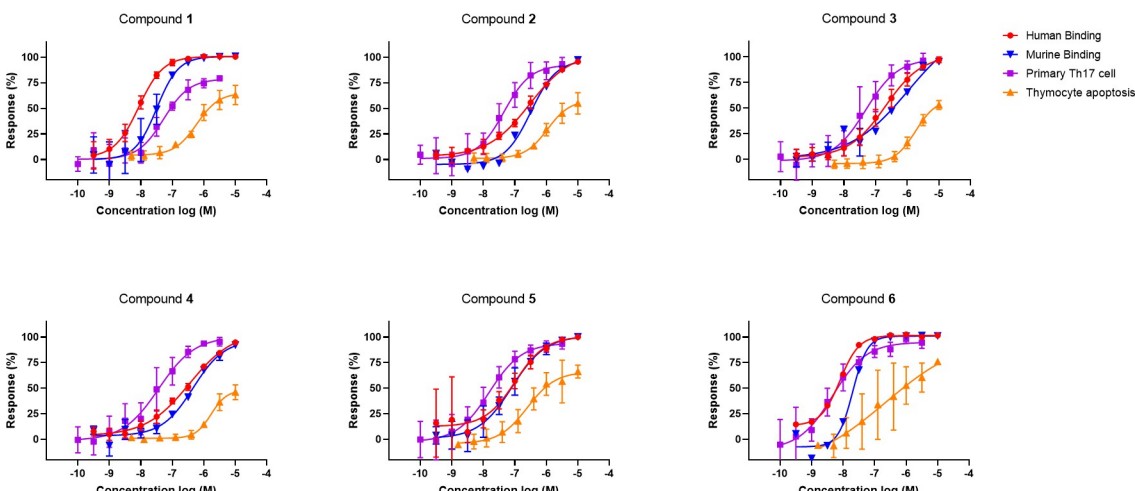

**Fig 1. Dose response curves for human and mouse ligand binding and functional assays.** Response vs concentration profile for compound **1**–**6** in the ligand binding assays (human = red, mouse = blue), the primary human Th17 cell assay (purple) and in the thymocyte apoptosis assay (orange). Error bars indicate standard deviation.

demonstrated a dose-dependent near complete inhibition of IL-17A production (93–99% vs control) with $pIC_{50}$ values in the range 7.2–8.2 (Table 1).

While the supernatants from the primary human Th17 cell assay were analysed for IL-17A protein levels, the cells in the corresponding wells were lysed and analysed for ATP content as an indicative measure of toxicity. This was done to increase confidence that the observed inhibition of IL-17A was neither due to a reduction in cell viability, nor proliferation. No reduction in ATP levels was observed for any of the compounds (S3 Fig).

**Evaluation of thymocyte apoptosis.** Since double positive thymocytes in the mouse require RORγt for their survival, we used this dependency to study the effect of RORγt inhibitors on thymocyte apoptosis. Mouse thymocytes were isolated, plated and incubated with compounds for 26 hours followed by AnnexinV and 7AAD staining to quantify apoptosis. The compounds all increased the basal rate of apoptosis and demonstrated a range of activities ($pEC_{50}$ 5.6–7.5) in the thymocyte apoptosis assay. Interestingly, the effect on murine thymocyte apoptosis required a significantly higher compound concentration than inhibition of IL-17A in human cells (Fig 1). Since binding potency is similar between the species, these data indicate a potential margin between the two effects.

**Target engagement through thymus involution *in vivo*.** After establishing an *in vitro* potency for the compounds on thymocyte apoptosis we set out to test this effect in a corresponding *in vivo* model. Compounds **3** and **4** were selected based on their favourable pharmacokinetic profiles. The *in vivo* profile of compound **5** has previously been published by our group [33].

Briefly, C57Bl/6NCrl mice were dosed with compounds twice daily for 2 days after which a single-cell suspension was prepared from each thymus. The cells were counted, stained and analysed by flow cytometry for $CD4^+CD8^+$ cell numbers and frequencies. We were able to demonstrate that the absolute number of double positive thymocytes was reduced in a dose dependent manner for compounds **3** and **4** (Fig 2A and 2B). This is in accordance with what was observed in the *in vitro* thymocyte apoptosis assay.

**Evaluation of systemic anti-inflammatory effects on psoriasis like skin inflammation in vivo.** Compound **3** was selected for further *in vivo* evaluation based on the stable predicted exposure levels over time (see Fig 2C and 2D).

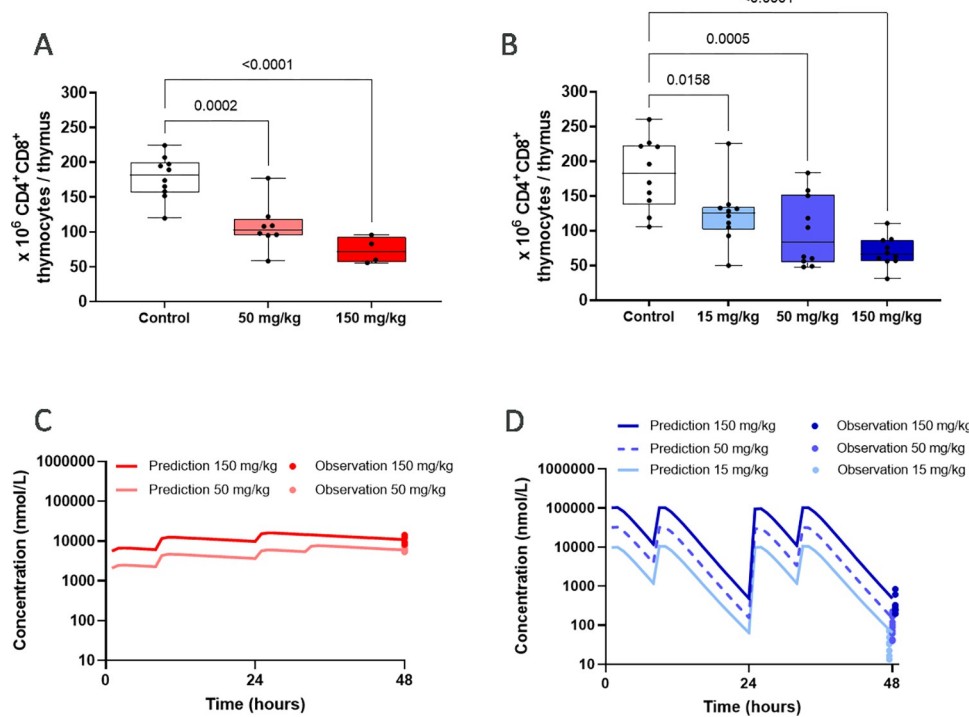

**Fig 2. Thymocyte numbers and exposure predictions and measurements after compound treatment in the thymus involution model.** A) Absolute numbers of CD4$^+$CD8$^+$ thymocytes in the thymus at termination after dosing BID for 48 hours with compound **3** B) and compound **4** in the thymus involution model *in vivo*. Significance levels: * (p ≤ 0.05), ** (p ≤ 0.01), *** (p ≤ 0.001). Exposure predictions (lines) and terminal concentrations (dots) in compound treated mice in the thymus involution model *in vivo* for C) compound **3** and D) compound **4**.

The anti-inflammatory effects of this compound were investigated in a psoriasis-like model of skin inflammation. In short, skin inflammation was induced by daily applications of IMQ cream for 7 days in the presence or absence of a RORγt inhibitor. Animals were dosed orally at 50mg/kg twice daily with compound **3**, which due to accumulation, led to an exposure at steady state similar to the high dose group (150 mg/kg) in the thymus involution study (S4 Fig). Animals were terminated on the morning of day 8, cells were harvested from the ears and cell populations were investigated by flow cytometry.

To assess the overall inflammation, cells were analysed for their capacity to produce IL-17A after restimulation *ex vivo* and frequencies of total IL-17A positive immune cells were shown to be reduced (p = 0.0002) (Fig 3A). Similarly, γδ T cells were found to produce significantly less IL-17A after compound treatment (p = 0.002) (Fig 3B) and their frequency was decreased (p = 0.006) (Fig 3C) indicating reduced recruitment and/or expansion of γδ T cells to the ear. Mean fluorescence intensity of IL-17A was also reduced for total IL-17A producing cells (p = 0.0004) as well as within the γδ T cell population (p = 0.02) in the compound treated group (S5 Fig). Taken together, compound treatment reduced the number of IL-17A positive cells and cells remaining positive for IL-17A expressed the protein at lower levels. Ear thickness was significantly decreased towards the end of the study (Fig 3D) indicating an anti-inflammatory effect of compound treatment (0.33 mm) as compared to vehicle (0.37 mm, p = 0.04) despite the continuous application of IMQ. A representative histogram of γδ T cells and individual data points for ear thickness are shown in S5 Fig.

**Inter-assay comparison.** To aid in the interpretation of these data we have evaluated the cross-species potencies and protein binding of the tested compounds.

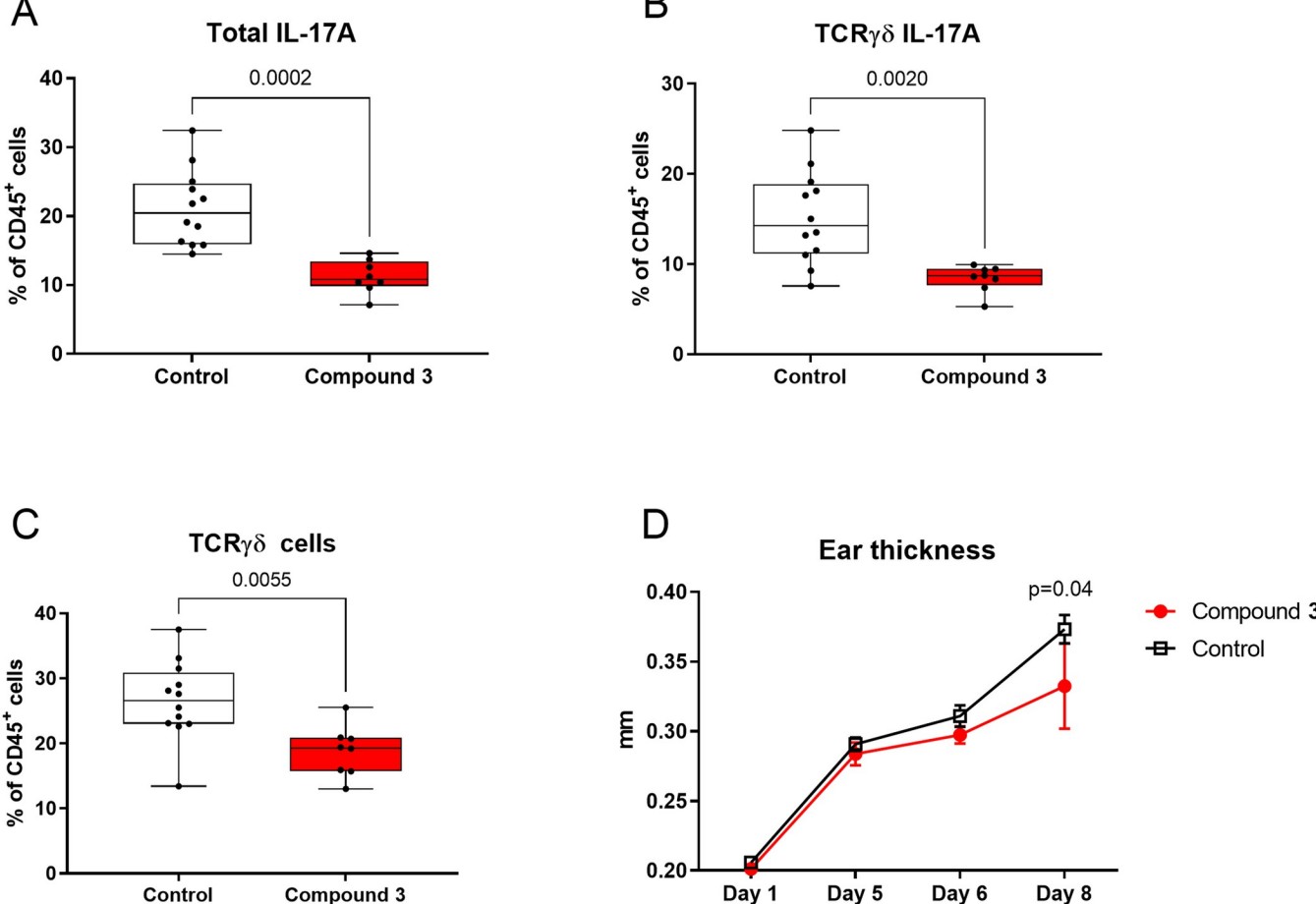

**Fig 3. Reduction of IL-17A and inflammatory readouts in the ear of IMQ treated mice after treatment with compound 3.** Cell populations in the ear of IMQ treated animals after treatment with compound **3**. A) Percentage of total IL-17A producing cells among CD45+ cells B) Percentage IL-17A producing cells among γδ T cells C) Percentage of γδ T cells among CD45$^+$ cells D) Ear thickness measured over the course of the IMQ induces skin inflammation model *in vivo*. Significance levels: * (p ≤ 0.05), ** (p ≤ 0.01), *** (p ≤ 0.001).

Among the six compounds, human and mouse inhibition constants (Ki) are generally similar (Fig 4A) with a trend towards higher potency in the human assay. This allows us to compare assay effects across species.

The desired effect is to inhibit IL-17A production in human Th17 cells. In this cell assay, for each concentration of compound, the effect on human IL-17A production was greater than the effect on human RORγt binding, except for Compound **1** (Fig 4B). However, this compound has a significantly higher protein bound fraction compared to the other five compounds (see Table 1), resulting in a reduction in the free compound concentration in assays with a higher protein content, such as the human IL-17A assay, potentially explaining the discrepancy.

In contrast, for each concentration of compound, the effect in the thymocyte apoptosis assay is consistently lower than the effect in the mouse ligand binding assay (Fig 4C).

For all compounds the effect is significantly lower in the thymocyte apoptosis assay compared to the human IL-17A production assay. Consequently, it is only when the inhibition of IL-17A production in the human cell is above 75% that the effect on mouse thymocyte apoptosis becomes significant (Fig 4D). The one compound that deviates somewhat from this

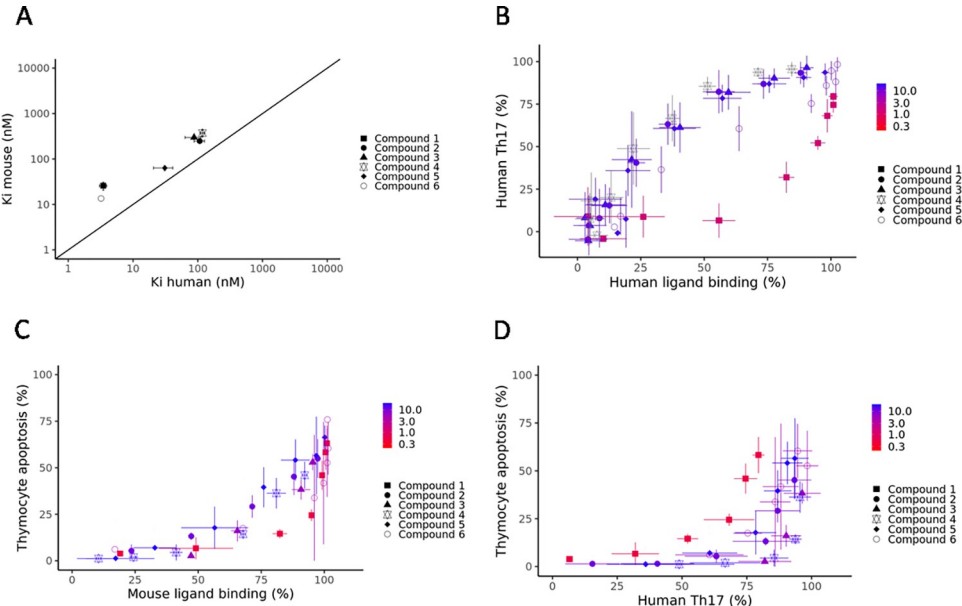

**Fig 4. Correlation of screening assays with functional readouts show separation of IL-17A and thymocyte apoptosis effects.** A) Comparison of human and mouse inhibition constant (Ki) for RORγt B) Comparison of percentage effect for the compounds in the human ligand binding assay vs observed inhibition in the primary human Th17 cell assay at corresponding concentrations. C) Comparison of percentage effect for compounds in the mouse ligand binding assay vs observed inhibition in the mouse thymocyte apoptosis assay at corresponding concentrations. D) Comparison of percentage effect for compounds in the human IL-17 assay vs observed inhibition in the mouse thymocyte apoptosis assay at corresponding concentrations. The colour scales depict the fraction unbound (%) in FBS.

behaviour (Compound **1**) shows high protein binding which we predict will lead to a greater loss of activity in the IL-17A assay than in the mouse thymocyte assay. We therefore conclude that these two effects require different levels of inhibition of RORγt with a greater degree of inhibition needed to affect thymocyte apoptosis.

## Discussion

Monoclonal antibodies directed against either IL-17A or IL-23 [37] have been demonstrated to be efficacious in both psoriasis patients and in spondyloarthropathies [38]. Due to the central role of RORγt in directing the cell-state and the production of multiple pro-inflammatory cytokines in several cell types, such as Th17, γδ T cells, ILC3 cells etc [39], inhibition may lead to additional clinical efficacy. Whilst clinical experience with anti-IL-17A and anti-IL-23 antibodies has not revealed significant safety concerns [40], the shorter half-life of a small molecule may allow for rapid treatment discontinuation, for example during clearance of a serious mycobacterial infection.

However, as we and others have published previously, an obstacle to clinical progression for the RORγt inhibitory mechanism may be the additional role played by RORγt during the maturation of thymocytes. In mice where the RORγt gene has been removed, either from conception [17] or when animals have reached adulthood [41], thymic lymphoma has been shown to develop in a large fraction of the animals [19]. Furthermore, scientists at Bristol Myers Squibb have reported thymic lymphomas in both rats and mice, but not cynomolgus monkey, following dosing with the RORγt inverse agonist, BMS-986251 [42]. The translation of this effect to humans is not clear as indicated by the fact that no lymphomas were reported in three kindreds bearing loss-of-function mutations in the *RORC* gene in children up to the age of 9

years [20] while mice develop thymic lymphomas within 4–5 months of losing Rorc [19,41]. Nevertheless, the identification of compounds with a preferential effect on cytokine production over thymocyte maturation would be desirable. However, in a subset of acute lymphoblastic leukemia originating in the early T precursor cell (ETP-ALL), enrichment in the IL-17 pathway has been described [43]. Furthermore, during early thymocyte maturation, in the DN1 stage, a subset of thymocytes upregulate the expression of Rorc [44] and these cells can, if dysregulated, also develop into ETP-ALL. In this situation it may be valuable to evaluate if inhibition of RORγt reduces the proliferation and expansion of such cells.

In this paper we present *in vitro* data arguing that a lower level of inhibition is required to reduce production of Th17 cytokines than is required to induce apoptosis in murine thymocytes.

We have evaluated six compounds with similar potencies in the mouse and the human ligand binding assays allowing comparison across species. We demonstrate an enhanced potency on inhibition of IL-17A production in primary human Th17-cells compared to the ligand binding assay. Also, the potency of cofactor displacement is close to the potency in the human Th17 cell assay suggesting that these two assays better represent the functional potency of the compounds than does the ligand-domain binding assay. This observation has been seen previously for other series of compounds in similar cell assay systems [45–48].

Contrasting with this finding, there is a consistent drop in potency between the ligand binding assay and the thymus apoptosis assay. Taken together this provides potential evidence that a greater inhibition of RORγt is required to affect thymocyte apoptosis than to reduce cytokine production *in vitro*.

Through the IMQ model we were able to demonstrate anti-inflammatory effects of RORγt inhibition in the ear skin of mice. It is worth to note that this is one of many animal models used to mimic part of the complex pathophysiology of psoriasis [49]. The model only weakly resembles human psoriasis but allows the evaluation of systemic anti-inflammatory effects targeting RORγt biology [50]. Upon oral treatment, the frequency of IL-17A producing cells was significantly reduced in IMQ treated ear skin and the remaining IL-17A+ cells had reduced protein levels after stimulation, consistent with an overall reduction of inflammation. In fact, a statistically significant reduction in ear thickness could be identified towards the end of the experiment compared to vehicle treated animals, despite the tissue edema. The anti-inflammatory effect of RORγt inhibition has been demonstrated previously on IMQ induced skin inflammation [47,48,51]. One paper describes that an effective reduction of IL-17A protein in IMQ treated ears [48] could be achieved at compound exposures corresponding to the *in vitro* $IC_{50}$ in a murine IL-17A inhibition cell assay (compound 66, free $C_{min}$). However, a higher concentration corresponding to $IC_{80}$ (free $C_{min}$) was needed for a statistically significant decrease of IMQ induced swelling [48]. There are similarities with our results, where high concentrations of compound **3** led to a clear reduction in IL-17A positive cells in IMQ treated ears, but a partial response on ear thickness day 8. Additionally, in our previous paper ([33] Compound **5**) we observed a larger effect from repeated daily dosing of 15 mg/kg in the IMQ model whereas bi-daily administration of 10 mg/kg did not have a significant effect on thymocyte numbers after two days. These two administration regimes gave very similar exposure profiles but also indicated a differential effect between IL-17A effector cell function and thymocyte apoptosis. Also in this study, the ear thickness was significantly but partially reduced [33]. One possible explanation for the partial effect is that continuous application of a TLR7 agonist (IMQ) is likely to drive a broader inflammatory response than just the IL-17 arm and hence appear more resistant to RORγt inhibition [52].

Nevertheless, RORγt inhibitors in clinical development have been terminated due to the potential safety risks. Even though the full outcome from these studies remains to be

published, it is likely that this is due to the challenge of establishing a large enough safety margin to thymic changes (reviewed in [53]). To date, both mice and rats have been exposed to RORγt inverse agonists and at higher doses this has resulted in the development of thymic lymphomas [42,54]. Furthermore, at lower doses, changes to the thymic cellular organization have been detected in both mouse and rat, as well as cynomolgus monkey [42].

Today there are no established means to differentiate between tolerable compound-induced thymic changes, and pre-neoplastic changes that may lead to the development of thymic lymphoma and ultimately T-cell acute lymphoblastic leukemia (T-ALL) during chronic treatment. This has been discussed by Haggerty et al based on the partly disclosed findings using BMS-986251 in cynomolgus monkey [42]. An additional complication of inhibition of RORgt has been suggested by Guo et al [55] where they have demonstrated a skewing of the TCRa gene rearrangement and hence limitations to the diversity of the T cell repertoire in mice. While skewing of the T cell repertoire is noteworthy, due to the potentially enhanced risk of developing autoimmune disorders and infection, the T cell leukopenia observed in humans carrying RORC biallelic loss of function mutations, is less pronounced than that observed in mice [20].

Hence, today, clinical progression of this class of inhibitor seems to be an unsurmountable hurdle due to the lack of means to predict and evaluate the safety profile with regards to development of thymic lymphoma and T-ALL, which is a clear limitation. However, our observation of the requirement for higher levels of inhibition of RORγt to induce thymus apoptosis may open an avenue towards the development of further novel compounds with increased differentiation between Th-17 cells and thymocytes. This might be achieved by a partial inhibitor of the nuclear receptor such that the level of inhibition needed to affect thymocytes is never reached. Other approaches may be through engaging tissue specific cofactors and one publication suggests a differential dependency between Th17 cells and thymocyte differentiation upon Src family kinases for the transcriptional activity of RORγt [56].

Additionally, mutations in the DNA-binding domain of RORγt have been suggested to impart a similar separation of effect between the thymus and effector cells [57], although currently there has been no publication of compounds raised against this part of the protein.

Other possibilities to separate the thymus apoptosis from effector cells have been suggested through the different conformations adapted by RORγt upon binding to its two different RORγt response elements. This conformational difference has been suggested to affect the cofactor recruitment and hence it might be possible to harness these findings in compound design [58]. To our knowledge, only one clinically tested compound has been suggested to inhibit effector cells but spare the thymus, IMU-935 [59], however this compound has been withdrawn from further clinical development [60]. This recently developed compound would have been valuable to evaluate in our experiments with the hope that it would separate the thymic effects from the inhibition of Th17 cells.

This study has a number of limitations. The lack of experimental data on human thymocytes prevents a direct comparison with Th-17 cells with the need to bridge potencies across species. However, the anatomical location of the thymus close to the heart and natural involution in adults significantly hampers access to fresh tissue which generally is only available from young children undergoing open heart surgery. A further limitation is the lack of establishing an *in vivo* margin between the positive effect of RORγt inhibition (reduction of IL-17A) and the unwanted effect of thymocyte apoptosis in the same animal experiment. This was not achievable due to the stress-sensitive nature of the thymus which undergoes rapid involution upon handling (e.g. application of IMQ cream and dosing of compounds) of the animals. Furthermore, the already occurring natural thymic involution would introduce further variability in thymus size in the adult animals needed in the IMQ experiment, both for the immune system to be fully matured and to minimize variability from growth related differences in size and

thickness of ear pinnas and cell numbers. This limitation did not apply to the experiments studying thymus effects as the animals were intentionally young to minimize effects from early thymic involution. A potential complication in measuring thymus cellularity of double positive thymocytes is premature thymic escape. This process has not been investigated in this work, however most of the literature has investigated premature thymic double positive escape in the context of infection or inflammation [61,62]. However, it is established by us and others that loss of [16,17] or inhibition of [54] RORγt has a direct and rapid effect on apoptosis. Since these animals are healthy and in a clean environment, we attribute the majority of the rapid loss of double positive thymocytes to apoptosis. An additional limitation is the artificiality of the thymocyte apoptosis assay where the thymic organization is disrupted thereby limiting the normal cytokine milieu and cellular cross talk (TCR stimulation). To circumvent this the thymocytes could have been stimulated in different ways. However, the enhanced apoptosis is supported in the thymus involution in vivo study where signaling and cross talk are intact.

Another limitation is the unknown level of inhibition of RORγt needed to achieve a clinically relevant effect on disease remission in human patients which would require longer clinical studies.

In this paper we present enhanced pharmaceutical insight for RORγt specific compounds with drug like properties. Furthermore, we were able to establish their anti-inflammatory effects as measured by cytokine production *in vitro* and measured through overall tissue inflammation *in vivo*. In addition, a margin between the *in vitro* inhibitory effect on IL-17A and enhanced thymocyte apoptosis was defined. Nevertheless, the translation of an *in vitro* margin to the separation of clinical efficacy from lymphoma risk in the clinic is unclear. Without the demonstration of compounds with a safe efficacy profile *in vivo*, or clear evidence that the mechanism by which lymphomas are induced in rodents is not relevant to humans, further development of RORγt inverse agonists for the treatment of Th17 related disease remains problematic.

## Supporting information

**S1 Fig. Flow cytometric gating strategy from ear tissue.** Single cells were selected followed by CD45+ live cells, TCRγδ intermediate TCRβ negative cells, further selected on CD3 and IL-17A using a histogram.
(TIF)

**S2 Fig. Compound structures.**
(TIF)

**S3 Fig. Representative data of cell toxicity as measured by terminal ATP levels in the primary human Th17 cell assay.**
(TIF)

**S4 Fig. Exposure prediction and terminal concentrations in the IMQ model.** Exposure prediction (line) and terminal concentrations (dots) in eight mice treated with compound **3** following 7 days twice-daily oral dosing with 50 mg/kg in the IMQ-induced skin inflammation model.
(TIF)

**S5 Fig. Flowcytometric parameters and ear thickness from the IMQ model.** A) Mean fluorescence intensity of IL-17A in CD45+ cells B) Mean fluorescence intensity of IL-17A in γδ T cells C) Histogram of IL-17A in γδ T cells, grey: Vehicle, in red: Compound **3** D) Ear thickness

over time with individual data points visualised.
(TIF)

**S1 File. Raw data.**
(XLSX)

## Acknowledgments

We wish to thank the AstraZeneca laboratory animal science unit for their excellent help in animal husbandry and care taking. We also would like to thank Pharmaceutical Sciences for their help in preparing compound formulations for the animal work. We thank Roine Olsson and Sarah Lever for help in compound design and synthesis.

## Author Contributions

**Conceptualization:** Mia Collins, Rikard Pehrson, Hanna Grindebacke, Agnes Leffler, Linda Yrlid, Frank Narjes, Antonio Llinas, Thomas G. Hansson, David Keeling, Johan Jirholt.

**Data curation:** Mia Collins, Rikard Pehrson, Hanna Grindebacke, Agnes Leffler, Marie Ramnegård, Helena Rannikmäe, Nina Krutrök, Linda Yrlid, Charlotte Pollard, Ian Dainty, Frank Narjes, Stefan von Berg, Antonio Llinas, Anna Malmberg, Jane McPheat, Eva Hansson, Elisabeth Bäck, Jenny Bernström.

**Formal analysis:** Mia Collins, Rikard Pehrson, Hanna Grindebacke, Agnes Leffler, Marie Ramnegård, Helena Rannikmäe, Nina Krutrök, Linda Yrlid, Charlotte Pollard, Ian Dainty, Frank Narjes, Stefan von Berg, Antonio Llinas, Anna Malmberg, Jane McPheat, Eva Hansson, Elisabeth Bäck, Jenny Bernström, Johan Jirholt.

**Investigation:** Hanna Grindebacke, Ian Dainty, Frank Narjes, Stefan von Berg, Antonio Llinas, Anna Malmberg, Jane McPheat, Eva Hansson, Elisabeth Bäck, Jenny Bernström, Thomas G. Hansson, David Keeling, Johan Jirholt.

**Methodology:** Mia Collins, Rikard Pehrson, Agnes Leffler, Marie Ramnegård, Helena Rannikmäe, Nina Krutrök, Linda Yrlid, Charlotte Pollard, Frank Narjes, Stefan von Berg, Anna Malmberg, Jane McPheat, Eva Hansson, Elisabeth Bäck, Jenny Bernström, Johan Jirholt.

**Project administration:** Thomas G. Hansson, David Keeling.

**Supervision:** Thomas G. Hansson, David Keeling, Johan Jirholt.

**Validation:** Mia Collins, Helena Rannikmäe, Nina Krutrök, Linda Yrlid, Charlotte Pollard, Stefan von Berg, Antonio Llinas, Anna Malmberg, Jane McPheat, Elisabeth Bäck.

**Visualization:** Mia Collins, Rikard Pehrson, Agnes Leffler, Marie Ramnegård, Helena Rannikmäe, Nina Krutrök, Ian Dainty, Frank Narjes, Stefan von Berg, Antonio Llinas, Johan Jirholt.

**Writing – original draft:** Mia Collins, Rikard Pehrson, Hanna Grindebacke, Agnes Leffler, Marie Ramnegård, Helena Rannikmäe, Nina Krutrök, Linda Yrlid, Charlotte Pollard, Ian Dainty, Frank Narjes, Stefan von Berg, Antonio Llinas, Anna Malmberg, Jane McPheat, Eva Hansson, Elisabeth Bäck, Jenny Bernström, Thomas G. Hansson, David Keeling, Johan Jirholt.

**Writing – review & editing:** Mia Collins, Rikard Pehrson, Hanna Grindebacke, Agnes Leffler, Marie Ramnegård, Helena Rannikmäe, Nina Krutrök, Linda Yrlid, Charlotte Pollard, Ian Dainty, Frank Narjes, Stefan von Berg, Antonio Llinas, Anna Malmberg, Jane McPheat, Eva

Hansson, Elisabeth Bäck, Jenny Bernström, Thomas G. Hansson, David Keeling, Johan Jirholt.

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
