## [Decision Letter · Decision Letter 0]

28 May 2024

PONE-D-24-01191RORγt inverse agonists demonstrating a margin between inhibition of IL-17A and thymocyte apoptosisPLOS ONE

Dear Dr. Collins,

Thank you for submitting your manuscript to PLOS ONE. After careful consideration, we feel that it has merit but does not fully meet PLOS ONE’s publication criteria as it currently stands. Therefore, we invite you to submit a revised version of the manuscript that addresses the points raised during the review process.

Both the reviewers raised several pertinent issues that need to be addressed in a proper way.

We look forward to receiving your revised manuscript.

Kind regards,

Subhasis Barik

Academic Editor

PLOS ONE

Journal Requirements:

[I have read the journal's policy and the authors of this manuscript have the following competing interests: All authors are or have been employees of AstraZeneca. Several of the authors are or have been shareholders of AstraZeneca.]. 

Additional Editor Comments:

Both the reviewers raised several pertinent issues that need to be addressed in a proper way.

Reviewers' comments:

Reviewer's Responses to Questions

**Comments to the Author**

1. Is the manuscript technically sound, and do the data support the conclusions?

Reviewer #1: Partly

Reviewer #2: Partly

2. Has the statistical analysis been performed appropriately and rigorously? 

Reviewer #1: Yes

Reviewer #2: Yes

3. Have the authors made all data underlying the findings in their manuscript fully available?

Reviewer #1: Yes

Reviewer #2: Yes

4. Is the manuscript presented in an intelligible fashion and written in standard English?

Reviewer #1: Yes

Reviewer #2: Yes

5. Review Comments to the Author

Reviewer #1: The manuscript "RORγt inverse agonists demonstrating a margin between inhibition of IL-17A and thymocyte apoptosis" investigates a fundamental duality in T cell functions; where targeting the key transcription factor that promotes Th17 cell functions in autoimmune diseases, may also lead to a defect in thymic T cell development. The work looks into the problem from a clinical perspective and finds a difference in dose for RORγt inverse agonists to exert these two opposite effects, which might be of therapeutic significance. Overall, the work has merit, but also has certain technical pitfalls. I have a few points for suggestions, as listed below.

a. The primary aim of this work is to uncouple CD4+CD8+ thymocyte apoptosis from Th17 cell activity modulation by the RORγt-targeted compounds. Since RORγt expression is thymus-specific, while RORγ is not expressed in the thymus to a considerable extent, the authors might have chosen a compound that selectively targets RORγ and not RORγt. This could have completely abolished any thymus-specific effect. The authors need to justify this.

b. IL-17 and other proinflammatory cytokine signalling pathways are highly active in early T cell progenitors and are associated with their leukemic potential (reference doi: 10.3389/fcell.2022.899752). Even though the authors have mentioned the implications of RORγt inhibition on thymic lymphoma in mice, they have not mentioned the potential effects of such therapies on the development of thymic T cell leukemia, which should be separately addressed as a limitation of the study. Similarly, the authors have also not mentioned the importance of RORγt on the selection of diverse TCRs in the thymus, which is a major hindrance to autoimmunity (reference doi: 10.1016/j.celrep.2016.11.073). The potential risks of developing T cells with autoimmune features by RORγt inhibition should also be discussed.

c. The ex vivo mouse thymocyte apoptosis assay in presence or absence of the compounds of interest has been performed without stimulating the thymocytes. However, apoptosis of thymocytes in the thymus during the selection checkpoints is dependent on a T cell receptor-mediated activation signal. The authors need to point out this fact and mention the need for assessing thymocyte apoptosis by these compounds upon ex vivo stimulation.

d. Even though the authors show significant success of compound 3 in reducing the levels of IL-17A as well as γδ T cells in the ear after imiquimod-induced skin inflammation, they might have also shown their levels in the systemic circulation (for example, in blood or spleen), to establish its role in moderating the systemic inflammation.

e. The authors have stated their inability to test their compound's efficacy in mediating thymocyte apoptosis during imiquimod-mediated skin inflammation. This could have been overcome by changing the dose of imiquimod which would have induced skin inflammation but with minimal damage to the thymus. Alternatively, IL-17A-dependent established inflammatory models which do not cause significant damage to the thymus (reference doi: 10.1111/j.1365-3083.2007.01923.x), might have been used. The authors need to discuss this issue.

f. Please replace "unspecific" with "non-specific" (Materials and methods).

Reviewer #2: Psoriasis and associated autoimmune manifestations are a set of chronic immunopathologies without plenty of available treatment options. Against such a backdrop, the authors have explored a unique side of anti-psoriatic medication, where a certain dose of an RORgammaT inverse agonist ameliorates psoriatic symptoms at the periphery without affecting T cell development. This is definitely of immense clinical importance, since defective thymopoiesis is often associated with several immunosuppressive therapies, such as corticosteroid therapy; unavoidably producing numerous side effects. While I do not find any major drawbacks in the rationale and design of the study, I have a few questions and suggestions towards the authors, as follows.

In figure 2, where the authors estimate the effects of their compounds on thymic DP cells in vivo, they have only measured the cell numbers and have shown a steady decrease in it with increasing concentration of their compound. How do they attribute this to be caused by apoptosis and not premature thymic escape of the DP thymocytes (de Meis et al, Journal of Parasitology Research, 2012) without specific assays conducted in vivo which indicate apoptosis (Annexin V binding, TUNEL, cleaved Caspase-3 measurement etc)? The correlation with their in vitro findings does not suffice for such an explanation.

The authors only look at the DP thymocyte number upon treatment with their compounds, and conclude that they are not affecting thymopoiesis to a great extent. However, several recent studies highlight the presence of crucial developmental events which, if perturbed, affect the functionality of the ensuing T cells upon maturation (Gamble et al, Nature Immunology, 2024; Bovolenta et al, PNAS, 2022) without taking a toll on their survival. Since RORgt has an important role in thymopoiesis, how do the authors ensure that their compounds are not affecting the epigenetic circuitry inside the developing thymocytes in any such way?

The methods are described in a slightly haphazard way. While there are plenty of details regarding how the authors validate the screened compounds in terms of their binding capacity, there is insufficient information about which compounds (or library) were deemed qualified for screening, and based on what criteria. The methodology related to the imiquimod-induced skin inflammation model is also written in a wayward manner. When is the compound applied, before or after imiquimod application?

The connection between RORgammaT manipulation and thymic lymphoma is very interesting. The authors should mention the risks of thymic T cell progenitor leukemia as well, since dysregulated IL-17 signaling and RORgammaT-dependent pathways are critically involved in the emergence of early T cell progenitor leukemia (Mukherjee et al, Frontiers in Cell & Developmental Biology, 2022).

In the discussion, the authors mention the variability in thymus size of 6-8 weeks old mice as a limitation of this study. Again, they mention the age of mice used in the thymic involution experiments as 5 weeks. They need to explain this discrepancy.

The authors test the efficacy of their compounds in an imiquimod-induced dermatitis model, which very weakly resembles psoriasis (Hawkes et al, Journal of investigative dermatology, 2017). This needs to be clearly mentioned in the discussion section.

6. PLOS authors have the option to publish the peer review history of their article (what does this mean?). If published, this will include your full peer review and any attached files.

Reviewer #1: No

Reviewer #2: No

---

## [Author Response · Author response to Decision Letter 0]

26 Aug 2024

Editor’s summary and main concerns: 

Thank you for submitting your manuscript to PLOS ONE. After careful consideration, we feel that it has merit but does not fully meet PLOS ONE’s publication criteria as it currently stands. Therefore, we invite you to submit a revised version of the manuscript that addresses the points raised during the review process.

Dear Editor

Thank you for the possibility to revise this manuscript. It is clear to us that the reviewers have spent substantial time and effort in reviewing our work in order to help us make improvements. 

We have formulated responses to each of the questions below and updated the manuscript accordingly.

We believe that the paper is improved from the feedback we have received and that the text is clearer and easier to read and understand.

Yours sincerely,

Dr Mia Collins

Review Comments to the Author

 Reviewer 1: 

Reviewer #1: The manuscript "RORγt inverse agonists demonstrating a margin between inhibition of IL-17A and thymocyte apoptosis" investigates a fundamental duality in T cell functions; where targeting the key transcription factor that promotes Th17 cell functions in autoimmune diseases, may also lead to a defect in thymic T cell development. The work looks into the problem from a clinical perspective and finds a difference in dose for RORγt inverse agonists to exert these two opposite effects, which might be of therapeutic significance. Overall, the work has merit, but also has certain technical pitfalls. I have a few points for suggestions, as listed below.

a. The primary aim of this work is to uncouple CD4+CD8+ thymocyte apoptosis from Th17 cell activity modulation by the RORγt-targeted compounds. Since RORγt expression is thymus-specific, while RORγ is not expressed in the thymus to a considerable extent, the authors might have chosen a compound that selectively targets RORγ and not RORγt. This could have completely abolished any thymus-specific effect. The authors need to justify this. 

Thank you for this comment, it is indeed a great suggestion and it would have been optimal to include such a compound. However, except for the compound referenced in the discussion of the manuscript (Reference 54 Polasek et al), we are not aware of any compound that distinguishes between the 2 isoforms and the referenced compound was not available at the time of the study. The protein sequence of the ligand binding domain is identical between the RORγ and RORγt isoforms, and isoform specificity is likely to be very hard to achieve. Furthermore, since both isoforms are expressed in Th17 cells, it is likely that it will be necessary to inhibit both RORγ and RORγt to achieve efficacy in peripheral T-cells. This data is stated in the introduction, rows 77-84, and we have added an additional clarification to accommodate the reviewers feedback rows 667-669.

b. IL-17 and other proinflammatory cytokine signalling pathways are highly active in early T cell progenitors and are associated with their leukemic potential (reference doi: 10.3389/fcell.2022.899752). Even though the authors have mentioned the implications of RORγt inhibition on thymic lymphoma in mice, they have not mentioned the potential effects of such therapies on the development of thymic T cell leukemia, which should be separately addressed as a limitation of the study. 

This is an important point, and we share these thoughts and concerns. We have now made it even more clear than before that this might be a problem in humans and we added to this discussion at rows 646-649. The following section now reads: 

“Hence, today, clinical progression of this class of inhibitor seems to be an unsurmountable hurdle due to the lack of means to predict and evaluate the safety profile with regards to development of thymic lymphoma and T-ALL, which is a clear limitation.”

Similarly, the authors have also not mentioned the importance of RORγt on the selection of diverse TCRs in the thymus, which is a major hindrance to autoimmunity (reference doi: 10.1016/j.celrep.2016.11.073). The potential risks of developing T cells with autoimmune features by RORγt inhibition should also be discussed. 

We would like to thank the reviewer for this important point. We have updated the manuscript to reflect this data and its potential consequences. Rows 638-644 which now reads: 

“An additional complication of inhibition of RORgt has been suggested by Guo et al [51] where they have demonstrated a skewing of the TCRa gene rearrangement and hence limitations to the diversity of the T cell repertoire in mice. While skewing of the T cell repertoire is noteworthy, due to the potentially enhanced risk of developing autoimmune disorders and infection, the T cell leukopenia observed in humans carrying RORC biallelic loss of function mutations, is less pronounced than that observed in mice [19].” 

c. The ex vivo mouse thymocyte apoptosis assay in presence or absence of the compounds of interest has been performed without stimulating the thymocytes. However, apoptosis of thymocytes in the thymus during the selection checkpoints is dependent on a T cell receptor-mediated activation signal. The authors need to point out this fact and mention the need for assessing thymocyte apoptosis by these compounds upon ex vivo stimulation. 

Thank you for this comment, it is important to bring up several aspects of this complicated biology. Thymocytes apoptose naturally in the thymus both due to an overly strong TCR engagement and a too week, or lack of engagement aka “death by neglect”. We have updated the text to reflect this and to make it more clear. Rows 67-69 now reads:

“In the mouse, thymocytes deficient in RORγt remain hyperproliferative but they also rapidly apoptose through the lack of RORγt-dependent Bcl-xL induced quiescence [16, 18].”

d. Even though the authors show significant success of compound 3 in reducing the levels of IL-17A as well as γδ T cells in the ear after imiquimod-induced skin inflammation, they might have also shown their levels in the systemic circulation (for example, in blood or spleen), to establish its role in moderating the systemic inflammation. 

This is a great suggestion and in hindsight we might have included this. The IMQ model mainly induces a local inflammation in the ear during the 8-day treatment period. The point we would like to make with this experiment is to demonstrate that through peroral drug administration we achieve systemic exposure at sufficient levels to block inflammation even in the periphery such as the ear pinnas. As we have demonstrated exposure in the blood, we have no doubt that we will also affect inflammation throughout the body.

e. The authors have stated their inability to test their compound's efficacy in mediating thymocyte apoptosis during imiquimod-mediated skin inflammation. This could have been overcome by changing the dose of imiquimod which would have induced skin inflammation but with minimal damage to the thymus.

Alternatively, IL-17A-dependent established inflammatory models which do not cause significant damage to the thymus (reference doi: 10.1111/j.1365-3083.2007.01923.x), might have been used. The authors need to discuss this issue. 

Thank you for these comments. While several processes affect thymocyte survival, we have not been able to find any data indicating that IMQ causes thymic damage. Ageing, however, comes with the involution of the thymus where the organ reduces in size with age, as well as its output of naïve T-cells. This process is variable in individuals and across the sexes. This variability hampers the possibility to use the thymic readouts in such models where ethical considerations dictate the usable age span of the experimental animals. Thymus involution sets in approximately at sexual maturation in mice-(6-8 weeks). Additionally, the daily application of IMQ creme and per oral compound administration induces some stress to the animals that affects the cellularity and weight of the animals’ thymae. In line with this reasoning, the suggested use of the dextran sulphate sodium-induced intestinal inflammation model is unlikely to aid in circumventing the stress, malaise and age dependent variability of the thymus size and cellularity. We have updated the section that highlights that this separation between thymic effects and effect on IMQ induced inflammation is important and subject to stress and age dependent variability. Rows 675-685 now read:

“A further limitation is the lack of establishing an in vivo margin between the positive effect of RORγt inhibition (reduction of IL-17A) and the unwanted effect of thymocyte apoptosis in the same animal experiment. This was not achievable due to the stress-sensitive nature of the thymus which undergoes rapid involution upon handling (e.g application of IMQ cream and dosing of compounds) of the animals. Furthermore, the early onset of natural thymic involution in mice around sexual maturity (6-8 weeks) would introduce further variability in thymus size in the adult animals needed in the IMQ experiment, both for the immune system to be fully matured and to minimize variability from growth related differences in size and thickness of ear pinnas and cell numbers. This limitation did not apply to the experiments studying thymus effects as the animals were intentionally young to ensure thymic involution had not yet started.”

f. Please replace "unspecific" with "non-specific" (Materials and methods). 

Thank you, this has been corrected.

Reviewer 2: 

Reviewer #2: Psoriasis and associated autoimmune manifestations are a set of chronic immunopathologies without plenty of available treatment options. Against such a backdrop, the authors have explored a unique side of anti-psoriatic medication, where a certain dose of an RORgammaT inverse agonist ameliorates psoriatic symptoms at the periphery without affecting T cell development. This is definitely of immense clinical importance, since defective thymopoiesis is often associated with several immunosuppressive therapies, such as corticosteroid therapy; unavoidably producing numerous side effects. While I do not find any major drawbacks in the rationale and design of the study, I have a few questions and suggestions towards the authors, as follows.

a. In figure 2, where the authors estimate the effects of their compounds on thymic DP cells in vivo, they have only measured the cell numbers and have shown a steady decrease in it with increasing concentration of their compound. How do they attribute this to be caused by apoptosis and not premature thymic escape of the DP thymocytes (de Meis et al, Journal of Parasitology Research, 2012) without specific assays conducted in vivo which indicate apoptosis (Annexin V binding, TUNEL, cleaved Caspase-3 measurement etc)? The correlation with their in vitro findings does not suffice for such an explanation. 

Thank you for this comment. We show in this paper that we induce apoptosis in freshly isolated primary thymocytes ex vivo through AnnexinV staining. While this does not constitute evidence for the same process taking place in vivo, it demonstrates that high levels of inhibition of RORgt leads to apoptosis in the same way as loss of function of RORgt does. That loss of RORgt induces apoptosis in vivo has been established previously, both through AnnexinV staining of thymocytes and TUNEL staining in thymic sections (reference 18, Ueda 2002). Based on these established findings we believe that we can conclude that we lose thymocytes primarily through apoptosis. We have added this reference to row 69 to make this clearer to the reader. Reviewer 1 had a similar comment and hence we have also made changes to the text to reflect this at rows 67-69.

“In the mouse, thymocytes deficient in RORγt remain hyperproliferative but they also rapidly apoptose through the lack of RORγt-dependent Bcl-xL induced quiescence [16, 18].”

b. The authors only look at the DP thymocyte number upon treatment with their compounds, and conclude that they are not affecting thymopoiesis to a great extent. However, several recent studies highlight the presence of crucial developmental events which, if perturbed, affect the functionality of the ensuing T cells upon maturation (Gamble et al, Nature Immunology, 2024; Bovolenta et al, PNAS, 2022) without taking a toll on their survival. 

Since RORgt has an important role in thymopoiesis, how do the authors ensure that their compounds are not affecting the epigenetic circuitry inside the developing thymocytes in any such way? 

This is a good point; however, the paper aims to suggest a potential for a window between the effect on mature effector T-cells and thymus apoptosis. It is likely that with a substantial inhibition of RORγ in the thymus of mice many perturbations to the normal T-cell development can be expected, including epigenetic effects and effects on TCRα splicing and recombination. These can be inferred from the observed sustained proliferation followed by rapid apoptosis vs a proliferative burst followed by quiescence and then more limited apoptosis. Additionally, a skewing of the TCR repertoire has been observed both in mice and man. Some of the underlying causes have been described, like the mentioned lack of Bcl-xL upregulation. If we can circumvent the induced apoptosis, then these other issues can be studied. This has not been accomplished in this work, but we have enhanced the discussion on potential consequences of inhibiting RORγ. There was a similar comment from Reviewer 1 and this was addressed in rows 638-644. 

“An additional complication of inhibition of RORgt has been suggested by Guo et al [51] where they have demonstrated a skewing of the TCRa gene rearrangement and hence limitations to the diversity of the T cell repertoire in mice. While skewing of the T cell repertoire is noteworthy, due to the potentially enhanced risk of developing autoimmune disorders and infection, the T cell leukopenia observed in humans carrying RORC biallelic loss of function mutations, is less pronounced than that observed in mice [19].” 

The methods are described in a slightly haphazard way. While there are plenty of details regarding how the authors validate the screened compounds in terms of their binding capacity, there is insufficient information about which compounds (or library) were deemed qualified for screening, and based on what criteria. 

Thank you for this comment. In this paper we have focused on describing biological effects of RORγt inhibition in developing thymocytes and peripheral mature T cells that are of general interest to the field. The chemical development of this series of RORγt inverse agonists have been described in three publications by us (see references 30-32) where a detailed account of the chemical evolution as well as the desired properties are presented. We have updated the introduction rows 100-101 to emphasize that information regarding the chemistry campaign is described in the three papers referenced.

The methodology related to the imiquimod-induced skin inflammation model is also written in a wayward manner. When is the compound applied, before or after imiquimod application? 

Thank you for pointing this out. This is an omission on our part and the text has been updated to clarify timing and order of manipulations. Rows 270-271 now reading:

“The application of Aldara cream took place in the morning after oral compound dosing.”

The connection between RORgammaT manipulation and thymic lymphoma is very interesting. The authors should mention the risks of thymic T cell progenitor leukemia as well, since dysregulated IL-17 signaling and RORgammaT-dependent pathways are critically involved in the emergence of early T cell progenitor leukemia (Mukherjee et al, Frontiers in Cell & Developmental Biology, 2022). 

Thanks for raising these points, it is indeed a very interesting paper and it is supported by other papers. Tarantini et al. Biomarker Research (2021) 9:89 https://doi.org/10.1186/s40364-021-00347-z

However, there is no mention of neither IL-17 nor RORγt in either of these papers as being active in ETP as stated by the reviewer. The current data supports that loss of RORγ or RORγt function in the mouse can lead

---

## [Decision Letter · Decision Letter 1]

17 Oct 2024

PONE-D-24-01191R1RORγt inverse agonists demonstrating a margin between inhibition of IL-17A and thymocyte apoptosisPLOS ONE

Dear Dr. Collins,

Thank you for submitting your manuscript to PLOS ONE. After careful consideration, we feel that it has merit but does not fully meet PLOS ONE’s publication criteria as it currently stands. Therefore, we invite you to submit a revised version of the manuscript that addresses the points raised by both the reviewers. Please submit your revised manuscript by Dec 01 2024 11:59PM. If you will need more time than this to complete your revisions, please reply to this message or contact the journal office at plosone@plos.org. Please include the following items when submitting your revised manuscript:A rebuttal letter that responds to each point raised by the academic editor and reviewer(s). You should upload this letter as a separate file labeled 'Response to Reviewers'.A marked-up copy of your manuscript that highlights changes made to the original version. You should upload this as a separate file labeled 'Revised Manuscript with Track Changes'.An unmarked version of your revised paper without tracked changes. You should upload this as a separate file labeled 'Manuscript'.

We look forward to receiving your revised manuscript.

Kind regards,

Subhasis Barik

Academic Editor

PLOS ONE

Additional Editor Comments:

Authors are requested to clarify the raised issues by both the reviewers

Reviewers' comments:

Reviewer's Responses to Questions

**Comments to the Author**

1. If the authors have adequately addressed your comments raised in a previous round of review and you feel that this manuscript is now acceptable for publication, you may indicate that here to bypass the “Comments to the Author” section, enter your conflict of interest statement in the “Confidential to Editor” section, and submit your "Accept" recommendation.

Reviewer #1: (No Response)

Reviewer #2: (No Response)

2. Is the manuscript technically sound, and do the data support the conclusions?

Reviewer #1: Yes

Reviewer #2: Partly

3. Has the statistical analysis been performed appropriately and rigorously? 

Reviewer #1: Yes

Reviewer #2: Yes

4. Have the authors made all data underlying the findings in their manuscript fully available?

Reviewer #1: Yes

Reviewer #2: Yes

5. Is the manuscript presented in an intelligible fashion and written in standard English?

Reviewer #1: Yes

Reviewer #2: Yes

6. Review Comments to the Author

Reviewer #1: The authors have addressed most of my concerns. I have a minor issue with their response to my comment no. c in the previous round of review. In their response as well as the modification in the manuscript, they do not mention the need for assessing thymocyte apoptosis in the presence of ex vivo stimulation. What they tried to convey through the modified text in the manuscript was not clear to me in this context. Indeed, thymocytes die in the thymus both due to excessively strong TCR stimulation or no stimulation at all. While they have analysed the effects of their compounds on the thymocytes in absence of any stimulation, it would be of greater value if they could assess the same in the presence of ex vivo stimuli.The authors need to give a more to-the-point explanation for this.

Reviewer #2: Collins et al have demonstrated formidable sincerity in addressing my comments. However, some of their responses could not directly address the particular queries raised by me.

Apoptosis and thymic escape of DP thymocytes are not mutually exclusive events [de Meis, Juliana et al. “Thymus atrophy and double-positive escape are common features in infectious diseases.” Journal of parasitology research vol. 2012 (2012): 574020. doi:10.1155/2012/574020; Démoulins, Thomas et al. “Reversible blockade of thymic output: an inherent part of TLR ligand-mediated immune response.” Journal of immunology (Baltimore, Md. : 1950) vol. 181,10 (2008): 6757-69. doi:10.4049/jimmunol.181.10.6757]. How do the authors, then, conclude that the loss of DP thymocytes does not involve premature DP escape as a potential reason? Simply referring to the fact that DP apoptosis is predominant during thymic involution does not suffice for this statement.

The authors state that the reference mentioned by me [Mukherjee, Soumyadeep et al. “In Silico Integration of Transcriptome and Interactome Predicts an ETP-ALL-Specific Transcriptional Footprint that Decodes its Developmental Propensity.” Frontiers in cell and developmental biology vol. 10 899752. 13 May. 2022, doi:10.3389/fcell.2022.899752] does not explicitly mention that ETP-ALL has high IL-17 activity. However, figure 2F in that article clearly shows a strong enrichment of IL-17 signaling in ETP-ALL. Furthermore, defined subsets of DN1 thymocytes (which also contribute to ETP-ALL) express a handsome amount of RORC transcript [Spidale, Nicholas A et al. “Interleukin-17-Producing γδ T Cells Originate from SOX13+ Progenitors that Are Independent of γδTCR Signaling.” Immunity vol. 49,5 (2018): 857-872.e5. doi:10.1016/j.immuni.2018.09.010], whereby it cannot be ruled out that RORgammaT inhibition might have some impact on ETP-ALL. I think the authors underestimated the implications of my comment in this context and it would be great if they highlight this aspect with due care.

6-8 weeks old mice are barely susceptible to age-induced thymic involution. However, natural age-associated thymic involution starts from approx. 4 weeks. Therefore, the authors’ claim regarding the optimality of their use of 5 weeks old mice for thymic involution experiments does not make much sense. In that case, they need to substantiate their claim by treating 0-2 weeks old pups with IMQ and check its effects on thymic involution. Nevertheless, I believe simply rephrasing their claim might be enough to avoid the confusion.

7. PLOS authors have the option to publish the peer review history of their article (what does this mean?). If published, this will include your full peer review and any attached files.

Reviewer #1: No

Reviewer #2: No

---

## [Author Response · Author response to Decision Letter 1]

25 Nov 2024

Editor’s summary and main concerns: 

Thank you for submitting your manuscript to PLOS ONE. After careful consideration, we feel that it has merit but does not fully meet PLOS ONE’s publication criteria as it currently stands. Therefore, we invite you to submit a revised version of the manuscript that addresses the points raised by both the reviewers.

Dear Editor

Thank you for the possibility to revise this manuscript again. We appreciate the reviewers’ time and effort with this, and we can see that the manuscript is now improved and that limitations with the study are now highlighted even more. 

We have formulated responses to each of the questions below and updated the manuscript accordingly. We hope that our responses will be satisfactory. 

Yours sincerely,

Dr Mia Collins

Review Comments to the Author

Reviewer 1: 

Reviewer #1: The authors have addressed most of my concerns. I have a minor issue with their response to my comment no. c in the previous round of review. In their response as well as the modification in the manuscript, they do not mention the need for assessing thymocyte apoptosis in the presence of ex vivo stimulation. What they tried to convey through the modified text in the manuscript was not clear to me in this context. Indeed, thymocytes die in the thymus both due to excessively strong TCR stimulation or no stimulation at all. While they have analysed the effects of their compounds on the thymocytes in absence of any stimulation, it would be of greater value if they could assess the same in the presence of ex vivo stimuli. The authors need to give a more to-the-point explanation for this.

Thank you for following up on this comment. 

In the normal situation DP thymocytes increase their expression of a fully formed TCRab receptor and begin to sample self peptides presented to them. However, DP thymocytes are dependent on Bcl-xL for their survival, the expression of which is governed by RORgt. Hence, blocking RORgt causes the loss of quiescence and cells begin to apoptose, effectively reducing the number of thymocytes expressing a fully formed TCRab receptor. This is an upstream event before thymocytes commit to becoming CD8+ Tc or CD4+ Th cells and leave the thymus as mature naive T-cells. It is therefore unlikely that exogenous CD3/CD28 stimulation or similar will affect this relatively short assay. In both mouse and human the number of mature peripheral T-cells is reduced in subjects where the RORgt protein function has been lost, but fully mature peripheral T-cells still exist. Apparently, a fraction of T-cells escape the thymus although RORgt has been lost, potentially depending on the timing or the strength of the positive selection signal (i.e. TCR stimulation) and this is an effect that we would suggest warrants further investigation. In this situation the reviewer’s suggestion is important to consider but not entirely straightforward to accomplish in this short assay. The assay is run as separated whole thymus cells, but the critical cellular organization and cytokine milieu is of course severely disrupted and quite difficult to mimic. As a follow up experiment to this assay, we do the in vivo experiment where similar effects are seen in an intact thymus with endogenous physiological TCR stimulation and selection.

Nonetheless, the limitations part of the discussion has been updated to reflect the reviewer’s important point. See rows: 701-706

“An additional limitation is the artificiality of the thymocyte apoptosis assay where the thymic organization is disrupted thereby limiting the normal cytokine milieu and cellular cross talk (TCR stimulation). To circumvent this the thymocytes could have been stimulated in different ways. However, the enhanced apoptosis is supported in the thymus involution in vivo study where signaling and cross talk are intact.”

Reviewer 2: 

Reviewer #2: Collins et al have demonstrated formidable sincerity in addressing my comments. However, some of their responses could not directly address the particular queries raised by me.

1. Apoptosis and thymic escape of DP thymocytes are not mutually exclusive events [de Meis, Juliana et al. “Thymus atrophy and double-positive escape are common features in infectious diseases.” Journal of parasitology research vol. 2012 (2012): 574020. doi:10.1155/2012/574020; Démoulins, Thomas et al. “Reversible blockade of thymic output: an inherent part of TLR ligand-mediated immune response.” Journal of immunology (Baltimore, Md. : 1950) vol. 181,10 (2008): 6757-69. doi:10.4049/jimmunol.181.10.6757]. How do the authors, then, conclude that the loss of DP thymocytes does not involve premature DP escape as a potential reason? Simply referring to the fact that DP apoptosis is predominant during thymic involution does not suffice for this statement.

Thank you for pointing out this important phenomenon. The reviewer correctly points out the lack of evidence to rule out premature thymic escape in our work. Having read the suggested papers and others, the published literature is limited in describing this in non-infectious or non-inflammatory situations, while being clear in establishing that loss of- or inhibition of- RORgt induces a direct and rapid DP thymocyte apoptosis. Given that our experimental animals are healthy and reside in a relatively clean environment, our conclusion is that the majority of the DP loss can be attributed to apoptosis. To highlight that the manuscript does not address this, we have updated the limitations part of the discussion to illuminate this shortcoming, see rows: 695-701

“A potential complication in measuring thymus cellularity of double positive thymocytes is premature thymic escape. This process has not been investigated in this work, however most of the literature has investigated premature thymic double positive escape in the context of infection or inflammation [61,62]. However, it is established by us and others that loss [16,17] or inhibition of [54] RORγt has a direct and rapid effect on apoptosis. Since these animals are healthy and in a clean environment, we attribute the majority of the rapid loss of double positive thymocytes to apoptosis.”

2. The authors state that the reference mentioned by me [Mukherjee, Soumyadeep et al. “In Silico Integration of Transcriptome and Interactome Predicts an ETP-ALL-Specific Transcriptional Footprint that Decodes its Developmental Propensity.” Frontiers in cell and developmental biology vol. 10 899752. 13 May. 2022, doi:10.3389/fcell.2022.899752] does not explicitly mention that ETP-ALL has high IL-17 activity. However, figure 2F in that article clearly shows a strong enrichment of IL-17 signaling in ETP-ALL. Furthermore, defined subsets of DN1 thymocytes (which also contribute to ETP-ALL) express a handsome amount of RORC transcript [Spidale, Nicholas A et al. “Interleukin-17-Producing γδ T Cells Originate from SOX13+ Progenitors that Are Independent of γδTCR Signaling.” Immunity vol. 49,5 (2018): 857-872.e5. doi:10.1016/j.immuni.2018.09.010], whereby it cannot be ruled out that RORgammaT inhibition might have some impact on ETP-ALL. I think the authors underestimated the implications of my comment in this context and it would be great if they highlight this aspect with due care.

We clearly underestimated the implications of the reviewer’s comments regarding ETP-ALL and IL-17 and ROR gamma and for this we are sorry. The reviewer is correct in highlighting that ETP-ALL are differentiated from non-ETP-ALL by an IL-17 signal and that DN1 cells can express ROR gamma. We have updated the text to reflect the involvement of this pathway in ETP-ALL, see rows: 590-596

“However, in a subset of acute lymphoblastic leukemia originating in the early T precursor cell (ETP-ALL), enrichment in the IL-17 pathway has been described [43]. Furthermore, during early thymocyte maturation, in the DN1 stage, a subset of thymocytes upregulate the expression of Rorc [44] and these cells can, if dysregulated, also develop into ETP-ALL. In this situation it may be valuable to evaluate if inhibition of RORyt reduces the proliferation and expansion of such cells.”

3. 6-8 weeks old mice are barely susceptible to age-induced thymic involution. However, natural age-associated thymic involution starts from approx. 4 weeks. Therefore, the authors’ claim regarding the optimality of their use of 5 weeks old mice for thymic involution experiments does not make much sense. In that case, they need to substantiate their claim by treating 0-2 weeks old pups with IMQ and check its effects on thymic involution. Nevertheless, I believe simply rephrasing their claim might be enough to avoid the confusion.

This is an important point that we apologize for overlooking. 

The mouse thymus involution indeed starts at 4 weeks of age. The most rapid decline in the mouse thymus starts after puberty and has a more rapid onset in males than females. In an experimental setting, stress has a far bigger impact on the thymus size at this time why precautions were taken to minimize stressors in the animal facility. We have reworked the text according to the reviewer’s suggestion, please see rows 246-251 which now reads:

“Female C57BL/6NCrl mice (Charles River, MA, USA) arrived at AstraZeneca at an age of 5 weeks +/- 2 days and were organized into groups and acclimatized for one week in order to reduce stress and variability of thymus weight. The mice in the experiment were intentionally young to reduce the impact of the normal process of thymic involution that starts at 4 weeks as this introduces further variability with regards to thymus size and cellularity. Additionally, age-related thymic involution is at its largest post puberty and less pronounced in females [34].”

And rows 689-695

“Furthermore, the already occurring natural thymic involution would introduce further variability in thymus size in the adult animals needed in the IMQ experiment, both for the immune system to be fully matured and to minimize variability from growth related differences in size and thickness of ear pinnas and cell numbers. This limitation did not apply to the experiments studying thymus effects as the animals were intentionally young to minimize effects from early thymic involution.”

---

## [Decision Letter · Decision Letter 2]

22 Dec 2024

RORγt inverse agonists demonstrating a margin between inhibition of IL-17A and thymocyte apoptosis

PONE-D-24-01191R2

Dear Dr. Collins,

We’re pleased to inform you that your manuscript has been judged scientifically suitable for publication and will be formally accepted for publication once it meets all outstanding technical requirements.

Kind regards,

Subhasis Barik

Academic Editor

PLOS ONE

Additional Editor Comments (optional):

Reviewers' comments:

Reviewer's Responses to Questions

**Comments to the Author**

1. If the authors have adequately addressed your comments raised in a previous round of review and you feel that this manuscript is now acceptable for publication, you may indicate that here to bypass the “Comments to the Author” section, enter your conflict of interest statement in the “Confidential to Editor” section, and submit your "Accept" recommendation.

Reviewer #1: All comments have been addressed

Reviewer #2: All comments have been addressed

2. Is the manuscript technically sound, and do the data support the conclusions?

Reviewer #1: Yes

Reviewer #2: Yes

3. Has the statistical analysis been performed appropriately and rigorously? 

Reviewer #1: Yes

Reviewer #2: Yes

4. Have the authors made all data underlying the findings in their manuscript fully available?

Reviewer #1: (No Response)

Reviewer #2: Yes

5. Is the manuscript presented in an intelligible fashion and written in standard English?

Reviewer #1: Yes

Reviewer #2: Yes

6. Review Comments to the Author

Reviewer #1: The authors have incorporated all the suggestions in the manuscript and have satisfactorily addressed all the comments.

Reviewer #2: My comments have been addressed. I do not have any more queries or comments regarding this manuscript.

7. PLOS authors have the option to publish the peer review history of their article (what does this mean?). If published, this will include your full peer review and any attached files.

Reviewer #1: No

Reviewer #2: No

---

## [Editor Report · Acceptance letter]

6 Jan 2025

PONE-D-24-01191R2 

PLOS ONE

Dear Dr. Collins, 

I'm pleased to inform you that your manuscript has been deemed suitable for publication in PLOS ONE. Congratulations! Your manuscript is now being handed over to our production team.

Kind regards, 

on behalf of

Dr. Subhasis Barik 

Academic Editor

PLOS ONE